# Utilization of Municipal Solid Waste Incineration (MSWIFA) in Geopolymer Concrete: A Study on Compressive Strength and Leaching Characteristics

**DOI:** 10.3390/ma17184609

**Published:** 2024-09-20

**Authors:** Qiyong Xu, Ning Shang, Jae Hac Ko

**Affiliations:** 1Shenzhen Engineering Laboratory for Eco-Efficient Recycled Materials, School of Environment and Energy, Peking University Shenzhen Graduate School, University Town, Xili, Nanshan District, Shenzhen 518055, China; qiyongxu@pkusz.edu.cn (Q.X.); shangning.cool@163.com (N.S.); 2Department of Environmental Engineering, Jeju National University, 102 Jejudaehak-ro, Jeju-si 63243, Jeju, Republic of Korea

**Keywords:** municipal solid waste incineration fly ash, washing pretreatment, heavy metal leachability, compressive strength, geopolymer

## Abstract

This study explores the utilization of municipal solid waste incineration fly ash (MSWIFA) in geopolymer concrete, focusing on compressive strength and heavy metal leachability. MSWIFA was sourced from a Shenzhen waste incineration plant and pretreated by washing to remove soluble salts. Geopolymer concrete was prepared incorporate with washed or unwashed MSWIFA and tested under different pH conditions (2.88, 4.20, and 10.0). Optimal compressive strength was achieved with a Si/Al ratio of 1.5, water/Na ratio of 10, and sand-binder ratio of 0.6. The washing pretreatment significantly enhanced compressive strength, particularly under alkaline conditions, with GP-WFA (washed MSWIFA) exhibiting a 49.6% increase in compressive strength, compared to a 21.3% increase in GP-FA (unwashed MSWIFA). Additionally, GP-WFA’s compressive strength reached 41.7 MPa, comparable to that of the control (GP-control) at 43.7 MPa. Leaching tests showed that acidic conditions (pH 2.88) promoted heavy metal leaching, which increased over the leaching time, while an alkaline environment significantly reduced the leachability of heavy metals. These findings highlight the potential of using washed MSWIFA in geopolymer concrete, promoting sustainable construction practices, particularly in alkaline conditions.

## 1. Introduction

The role of municipal solid waste (MSW) incineration in global MSW management is expected to grow gradually due to an increase in waste generation and limited landfill space. Municipal solid waste incineration fly ash (MSWIFA) is a byproduct of the incineration process used to reduce the volume of municipal solid waste effectively. MSWIFA is categorized as hazardous due to the presence of toxic heavy metals, such as lead (Pb), cadmium (Cd), zinc (Zn), copper (Cu), chromium (Cr), and nickel (Ni) [1,2,3]. The disposal of MSWIFA presents significant environmental and health risks, necessitating the development of effective and safe utilization methods [4].

Utilizing MSWIFA in construction materials has been widely researched, with geopolymer cement emerging as a promising alternative to traditional Portland cement [5,6,7,8,9]. Since Joseph Davidovits, who recognized pyramid theory [10], established the foundation of geopolymer science [11], the geopolymer research field has continued to provide environmentally beneficial building materials. Geopolymer technology, which can use SiO_2_-Al_2_O_3_-bearing industrial wastes such as construction and demolition debris (C&D) [12,13,14], coal fly ash [15,16], sludges [17,18], furnace slag [19,20,21], and mining waste [22] has gained attention as an alternative to ordinary Portland cement (OPC) concrete [23,24,25,26,27,28,29,30,31,32]. In addition to the use of industrial byproducts for geopolymer concrete production, H.M. and Unnikrishnan [33] reviewed recent studies and concluded that the utilization of agricultural byproducts like rice husk ash, corncob ash, and sugarcane bagasse ash, for geopolymer concrete could be an option to produce sustainable construction materials. Geopolymer is a synthetic alkali aluminosilicate material formed by the reaction of a solid aluminosilicate with a highly concentrated aqueous alkali hydroxide or silicate solution [34]. The advantages of geopolymer concrete over OPC concrete include the ability to utilize industrial waste for high-performance concrete and a significant reduction in CO_2_ emissions [35,36,37].

In the presence of alkali, the active silicon and aluminum components in fly ash and other silico-aluminates form a stable network structure of [SiO4]^4−^ and [AlO4]^5−^ [38]. The basic structural units of this network include silico-aluminum chains (PS), silico-alumino-silico chains (PSS), and silico-alumino-di-silico chains (PSDS). The conceptual mechanism of geopolymerization involves dissolution, migration, gelation, reorganization, polymerization, and hardening as a solid aluminosilicate source transforms into a synthetic alkali aluminosilicate [39]. Depending on the raw material, alkali activator, and processing conditions, geopolymer can exhibit a wide variety of properties, such as a high compressive strength, low shrinkage, variable setting times, acid resistance, fire resistance, and low thermal conductivity [40,41,42,43,44,45]. Amorphous geopolymers are generally obtained at curing temperatures ranging from 20 to 90 °C, while crystalline ones form in autoclaves at 150–200 °C [46]. Also, the combination of different aggregates can impact the properties of geopolymer concrete [47].

Two types of cementitious materials can be produced by alkaline activation based on Si and Ca or Si and Al [48]. These alkali-activated products, which are potential candidates for alkaline cements with optimal binding properties, may have low loss on ignition (LOI < 5%), Fe_2_O_3_ content (<10%), reactive silica between 40 and 50%, a high percentage of particles with size lower than 45 µm, and a high content of the vitreous phase [48]. The contents of SiO_2_ in MSWIFA are typically low, not fulfilling the proper silica content. Therefore, an additional supplement for the Si source like metakaolin is needed.

Recent research has shown the potential of geopolymer technology to immobilize hazardous components in industrial byproducts, such as MSWIFA. For instance, Phair and Van Deventer [49] demonstrated that geopolymer matrices could effectively encapsulate heavy metals, reducing their leaching potential under acidic conditions. Similarly, Zhang et al. [50] reported that the incorporation of MSWIFA in geopolymer cement improved its compressive strength and durability compared to traditional cement.

Earlier studies on MSWIFA highlighted its high heavy metal content, necessitating pretreatment to make it more suitable for construction materials [51,52,53,54]. Washing pretreatment has been identified as an effective method to remove soluble salts, as demonstrated by Tian, Themelis and Bourtsalas [53] and Wang et al. [55]. The effect of washed MSWIFA on geopolymer cement with a higher compressive strength and lower heavy metal leaching compared to untreated MSWIFA has been reported [54,56,57]. This pretreatment method enhances the environmental safety and mechanical performance of MSWIFA, making it more suitable for use in construction materials such as geopolymer cement [57,58,59].

Despite the exclusive studies conducted in geopolymer concrete incorporated with MSWIFA, knowledge of the comprehensive effects of washing pretreatment, pH conditions, leachability, and strength development on geopolymer concrete remains scarce. Therefore, this study aims to assess the mechanical performance of geopolymer concrete incorporating washed MSWIFA and examine the leaching behavior of heavy metals from geopolymer concrete under various environmental conditions (pH 2.88, 4.20, and 10.0).

## 2. Materials and Methods

### 2.1. MSWIFA Preparation

MSWIFA was obtained from a waste incineration power plant in Shenzhen, China. As a pretreatment, MSWIFA was washed twice using tap water. MSWIFA was mixed with water in a ratio of 1:10 (MSWIFA to water) and stirred thoroughly to dissolve the soluble components. The mixture was allowed to settle for 12 h, and the supernatant containing dissolved salts and impurities was decanted. The washing process was repeated to ensure maximum removal. The washed MSWIFA was then dried at 105 °C to remove excess moisture before being used in the preparation of geopolymer cement. With the washing treatment, the reduction in MSWIFA components such as chlorides, sulfates, and heavy metals was determined. In addition, the weight loss after washing was measured.

### 2.2. Preparation of Geopolymer Concrete

For the geopolymer mortar, metakaolin (Hunan Chaopai Technology Co., Ltd., Changsha, Hunan, China), ISO standard sand (Xiamen ISO Standard Sand Co., Ltd., Xiamen, Fujian, China), and sodium silicate (Na_2_SiO_3_·9H_2_O) as an alkali activator were used. Deionized water was used for preparing alkali activators. Metakaolin served as the primary aluminosilicate source for geopolymerization. The standard sand was used to prepare the geopolymer mortar. MSWIFA was added to replace some portion of the metakaolin in the geopolymer mortar. The sample name 0% MSWIFA replacement (GP-control) indicates that no MSWIFA was used, while GP-FA means 20% of the metakaolin was replaced by MSWIFA and GP-WFA indicates 20% of the metakaolin was replaced by washed MSWIFA.

The geopolymer concrete were prepared by mixing metakaolin, MSWIFA (washed or unwashed), standard sand, and alkali activators using a cement mortar mixer (JJ-5 Type, Guangzhou Baiyun Bolai Highway Construction Equipment Co., Ltd., Guangzhou, Guangdong, China) at 15 RPM. Weighed amounts of metakaolin, MSWIFA, and standard sand were mixed in a cement mortar mixer for 5 min to ensure uniform distribution. The calculated amount of sodium silicate and sodium hydroxide solutions was then added and mixing continued for an additional 2 min to obtain a homogeneous mixture. The mixture was transferred to 50 × 50 × 50 mm molds and compacted to remove air voids and ensure a dense structure. The specimens were cured at room temperature (20 ± 1 °C) for 24 h. After initial curing, the specimens were demolded and placed in a standard curing chamber at 20 ± 1 °C with a relative humidity of ≥90% for 7 days to allow additional geopolymerization and strength development. Figure 1 presents the process of MSWIFA geopolymer concrete production.

### 2.3. Orthogonal Experiment for Geopolymer Concrete

The orthogonal experimental design was chosen to systematically investigate the influence of multiple factors on the compressive strength of geopolymer concrete in this study. An orthogonal experiment was designed to optimize the mix proportions for geopolymer concrete. The experiment evaluated the effects of various parameters including the Si/Al ratio, water/Na ratio, sand-binder ratio, and the addition of Ca(OH)_2_ on the compressive strength of geopolymer concretes. The parameters and their levels with the compressive strength of the geopolymer are shown in Table 1.

### 2.4. Compressive Strength Test

The compressive strength of the geopolymer concrete was tested after 7 days of curing using a universal testing machine. Each sample was subjected to a gradually increasing load until failure, and the maximum load at failure was recorded. The compressive strength (σ) was calculated using Formula (1).
(1)$$Compressive\;Strength\left(\sigma ,MPa\right)=\frac{Maximum\;Load\;for\;failure\;(10^{6}N)}{Cross-section\;Area\;(m^{2})}$$

### 2.5. Long-Term Leaching Test

The leaching characteristics of the cubes were evaluated by immersing the samples in deionized water and periodically analyzing the leachate for heavy metal content. The leaching tests were conducted under different pH conditions (2.88, 4.20, and 10.0) to simulate various environmental scenarios. The pH levels were selected based on the pH of SPLP simulating acid rain conditions (pH 4.20), the pH of TCLP (pH 2.88) representing acidic environments, and pH 10.0 representing alkaline environments. Each batch of cubes was placed in three identical containers with sufficient volume, and deionized water was added based on a liquid-solid ratio of 10 mL/g. The pH was adjusted daily using HNO_3_ solution to ensure the pH of the leachate in the three containers remained around 2.88, 4.20, and 10.0. pH was measured using a pH meter (FiveEasy Plus, Mettler Toledo, Greifensee, Zürich, Switzerland). The solution in each container was sampled every 28 days to measure the concentrations of heavy metals (Zn, Pb, Cu, Cd, Cr, and Ni). Simultaneously, the corresponding cubes were taken out to measure their compressive strengths. After removing the corresponding geopolymer concretes from the leaching containers, an equal volume of leachate was taken out to maintain the liquid–solid ratio of 10 mL/g in each container.

### 2.6. Analytical Method

An X-ray fluorescence spectrometer (XRF, OPTIM’X WDXRF, Thermo Fisher Scientific, Waltham, MA, USA) was used to measure the inorganic element composition of MSWIFA. pH was determined using a pH meter (Sartorius PB-10). A scanning electron microscope (SEM, Supra^®^55, Carl Zeiss, Oberkochen, Baden-Württemberg, Germany) was used for the morphology analysis of the geopolymer concrete. A microwave digestion apparatus was used for measuring the heavy metal content in MSWIFA. Inductively coupled plasma mass spectrometry (ICP-MS, XSERIES 2, Thermo Fisher Scientific, Waltham, MA, USA) was used to determine the heavy metal concentration in the leachate. A compressive strength tester (Shanghai Hengyi Precision Instrument Co., Ltd., HY(YE)-100008, Shanghai, China) was used to determine the compressive strength of the geopolymer concrete specimens.

## 3. Results and Discussion

### 3.1. Orthogonal Experiment Range Analysis

The range analysis of the orthogonal experiment results is shown in Table 2. From the range analysis, the factors affecting the compressive strength of geopolymer concrete in order of significance are Si/Al ratio > water/Na ratio > sand-binder ratio > Ca(OH)_2_ addition. The optimal level for each parameter is identified as a Si/Al ratio of 1.5, water/Na ratio of 10, sand-binder ratio of 0.6, and Ca(OH)_2_ addition of 0. The Si/Al ratio significantly influenced the compressive strength of the geopolymer concrete. The optimal ratio of 1.5 resulted in the highest compressive strength (37.03 MPa). Higher ratios (e.g., 2.3) led to lower strengths, indicating that maintaining an optimal balance of silicon and aluminum is crucial for achieving high compressive strength. This is because the proportion of Si-O-Si bonds increases with the increase in Si/Al, and the bond strength of Si-O-Si is higher than that of Si-O-Al. Liu et al. [60] revealed that the optimum Si/Al ratio of 1.5 to 2.5 was identified through literature review. The water/Na ratio impacted both the workability and mechanical properties of the geopolymer concrete. A ratio of 10 was found to be optimal, providing sufficient water for the geopolymerization process without compromising the structural integrity of the concrete. The sand-binder ratio affects the texture and compressive strength of the geopolymer concrete. A ratio of 0.6 was identified as optimal, ensuring enough binder to maintain the structure and strength of the concrete without making it too brittle. The addition of Ca(OH)_2_ had the least impact on the compressive strength. The minimal difference in strength across its levels (range of 2.49 MPa) suggests that its inclusion may not be necessary, especially when considering cost and complexity.

### 3.2. The Effects of Washing on MSWIFA Characteristics

Table 3 shows the chemical composition of unwashed and washed MSWIFA and metakaolin. The results indicated that soluble ions like Cl, Na, and K were significantly removed from MSWIFA by washing. The washing process led to a substantial weight reduction in MSWIFA, approximately 53% of the original weight, due to the removal of soluble components and non-settlement material. As shown in Table 3, the main components of metakaolin are active SiO_2_ and Al_2_O_3_, accounting for 51.4% and 45.4%, respectively. However, CaO was a main component of MSWIFA, occupying 38.9%. Cl, K_2_O, and Na_2_O also presented relatively high levels in unwashed MSWIFA at 9.9%, 6.5%, and 6.9%, respectively. MSWIFA washing increased the CaO content to 48.2% but decreased the content of Cl, K_2_O, and Na_2_O to 1.2%, 0.7%, and 1.1%, respectively. The weight loss of MSWIFA also influenced its heavy metal concentrations as shown in Table 4. The heavy metal concentrations of washed MSWIFA were higher than that of unwashed MSWIFA except those of Pb and Ni. Wang et al. [61] also reported that washing could increase the heavy metal content in fly ash. However, the increased concentrations of all heavy metals were not proportional to the mass loss of washed MSWIFA, indicating that the concentrations likely depended on the nature of each heavy metal. The decreased concentration of Pb and Ni might be due to the loss of Pb and Ni during the washing process.

### 3.3. Surface Characteristics of Geopolymer in Different Leaching Conditions

Figure 2 and Figure 3 show the images of GP-FA and GP-WFA, before and after the 16-week leaching at different pH levels. At pH 2.88, the geopolymer concrete showed varying degrees of cracks on the surface, and the edges started to show indistinct features due to corrosion (Figure 3b). At pH 4.20 and 10.0, no significant cracks appeared on the surface of the geopolymer concrete. This indicated that during the 16-week leaching process at pH 2.88, the leachate had a significant corrosive effect on the surface of the geopolymer.

### 3.4. Compressive Strength Development

#### 3.4.1. Compressive Strength Development of GP-Control, GP-FA and GP-WFA

The compressive strength (σ) development of GP-control, GP-FA, and GP-WFA cured in the standard curing chamber for 16 weeks is shown in Figure 4. The σ of GP-control developed gradually from 31.6 MPa to 43.7 MPa for 12 weeks, representing a 38.3% increase. However, a slight decrease of 8.9% was found in the 16th week, bringing the compressive strength down to 39.8 MPa. The σ of GP-FA developed quickly for the first 4 weeks (a 21.3% increase from its initial strength), but after the 8th week it stabilized between 27.4 MPa and 33.3 MPa. In contrast, GP-WFA displayed slower early strength development, compared to GP-FA. The slow start with GP-WFA may be due to the difficulties of homogenous mixing caused by faster reactions in the early stage [62]. However, its compressive strength steadily increased to 41.7 MPa by the 16th week, achieving an overall increase of 49.6% from its initial value. This significant improvement demonstrates the positive impact of washing MSWIFA on enhancing the compressive strength of the geopolymer. By the 16th week, the compressive strength of GP-WFA was comparable to that of GP-control, indicating that the washing pretreatment enhanced the material’s mechanical properties.

#### 3.4.2. Compressive Strength Development in Different pH Conditions

The compressive strength of GP-control, GP-FA, and GP-WFA under different pH conditions was also evaluated over 16 weeks (Figure 5). In acidic conditions (pH 2.88), the compressive strength of GP-control ranged from 31.0 MPa to 36.6 MPa, showing an overall increase of 18.1%. However, the strength began to decline after 12 weeks, with a 13.4% decrease by Week 16, indicating the destructive effects of the acidic environment on the geopolymer’s structure. In acidic environments (pH 2.88), the increased solubility of metal hydroxides and the dissolution of the aluminosilicate network in the geopolymer matrix led to the fracture of Si-O-Al bonds [63,64]. The development of fractures on the surface of the geopolymer increased the acid attack on the surface leading to an acceleration in the degradation of the geopolymer strength [63].

At pH 4.2, the compressive strength of GP-control remained more stable, increasing by 18.4% to reach 46.8 MPa by the 16th week. Similarly, in alkaline conditions (pH 10.0), the compressive strength increased by 11.5%, peaking at 44.9 MPa by the 12th week. This suggests that mildly acidic and alkaline environments are more favorable for the long-term stability of the geopolymer. At pH 4.2, the σ of GP-control maintained strength, likely because the presence of residual alkaline activators helps to buffer the mild acidic environment, slowing the degradation process [63].

For GP-FA, the compressive strength fluctuated between 22.9 MPa and 32.7 MPa, with a noticeable decline at pH 2.88 after the 8th week, showing a 29.9% decrease. The variation was less pronounced at pH 4.2 and 10.0, where the strength increased slightly by 14.6% and 10.1%, respectively. These fluctuations suggest that the unwashed MSWIFA in GP-FA was more susceptible to environmental degradation. Multiple factors likely affect the σ development of the geopolymer, including the Ca content, acidic intensity, impurities, the release of soluble ions, and the geopolymer’s buffering capacity. As shown in Section 3.2, the proportions of elements such as Cl, Na, and K in MSWIFA significantly decreased, while the proportion of Ca increased substantially after MSWIFA-washing. For geopolymer concrete, a high Cl content can significantly slow down the hardening rate of the geopolymer gel and reduce the σ of the geopolymer concrete by disrupting the continuity of the geopolymer colloidal structure [65]. In contrast, Ca can provide more reactive sites, leading to gel formation and hardening of the geopolymer concrete [57]. Therefore, a higher Ca content could result in the rapid hardening of the geopolymer concrete and a higher compressive strength [66].

In comparison, GP-WFA exhibited a more consistent performance, especially at pH 4.2 and 10.0, where the compressive strength remained relatively stable, with increases of 14.9% and 12.5%, respectively, over the 16-week period. However, at pH 2.88, the strength declined by 36.2% after reaching a peak of 40.1 MPa at the 8th week. This indicates that while washed MSWIFA enhances the geopolymer’s strength, the material is still vulnerable to degradation in highly acidic environments. Bie, Yin and Chen [37] indicated that the high pH environment promoted secondary polymerization to form more C-S-H bonds, enhancing the structural integrity and reducing the rate of leaching and degradation.

### 3.5. Leaching Characteristics under Different pH Conditions

#### 3.5.1. Zn Leachability under Different pH Conditions

The leachate analysis showed a high concentration of Zn in the highly acidic environment (pH 2.88) with washed MSWIFA as shown in Figure 6. Zn leachability was significantly higher in strong acidic environments due to the increased solubility of zinc compounds. The Zn leachability increased with extended contact time with the leaching solution. The leaching concentration of Zn in GP-FA was 52.6 µg/L in the 4th week but increased to 837.9 µg/L in the 16th week at pH 2.88. This indicates significant Zn leaching not only due to the breakdown of the geopolymer matrix but also due to the solubility of Zn under strong acidic conditions. The washing pretreatment significantly increased Zn leachability under pH 2.88 conditions. The leaching concentration of Zn in GP-WFA was 1525 µg/L at the 16th week, demonstrating a substantial increase in Zn leachability compared to unwashed MSWIFA (838 µg/L). This phenomenon is likely due to the Zn enrichment in washed MSWIFA (9419 mg/kg) compared to unwashed MSWIFA (6096 mg/kg) and implies that Zn concentration and pH conditions were dominant factors affecting the leachability of Zn in the strong acidic environment. In addition, acidic environments accelerate the breakdown of the geopolymer structure, increasing Zn solubility and mobility [64,67,68]. Zn leachability was lower in alkaline conditions, likely due to the formation of insoluble Zn(OH)_2_ precipitates [69,70]. At a mildly acidic pH of 4.2 and in an alkaline environment (pH 10), Zn leaching was less severe compared to pH 2.88, but still notable. The Zn concentration in GP-FA ranged from 17.1 µg/L to 59.6 µg/L at pH 4.2 and from 5.5 µg/L to 47.5 µg/L at pH 10. The Zn concentration in GP-WFA was higher than that in GP-FA at pH 4.2 but comparable to that in GP-FA at pH 10. Increasing the exposure time did not promote Zn leaching, reflecting the reduced solubility of Zn at pH 10. GP-control at all tested pH ranges showed a lower leachability compared to GP-FA and GP-WFA due to the low Zn content. The same phenomenon was observed in other metals’ leaching with GP-control.

#### 3.5.2. Pb Leachability under Different pH Conditions

In the highly acidic environment (pH 2.88), the Pb concentration in the leachate from GP-FA was relatively high (Figure 7). Initial measurements showed a concentration of approximately 13.0 µg/L. Over time, the concentration increased further to 322.7 µg/L at the 16th week. The washing pretreatment did not improve the Pb leachability. The initial concentration in the leachate was around 91.5 µg/L, which gradually increased to 341.4 µg/L over the experimental period. This is likely due to the significant corrosive effect of the strongly acidic environment on the geopolymer structure, enhancing the leachability of Pb. At pH 4.2, Pb leaching was substantially lower compared to pH 2.88. The initial concentration was approximately 1.3 µg/L, increasing to 2.5 µg/L from GP-FA over time. The washing pretreatment showed no significant change in Pb leachability at pH 4.22. The initial Pb concentration was around 1.4 µg/L, stabilizing at approximately 3.0 µg/L over the experimental period. At pH 10, the Pb leaching of GP-FA and GP-WFA was lower. The initial concentration of GP-FA and GP-WFA was around 1.7 µg/L and 1.1 µg/L, which reduced to 1.3 µg/L and 0.8 µg/L over time, respectively. Lower Pb leachability in alkaline conditions is due to the formation of insoluble Pb(OH)_2_ and other lead hydroxide complexes. Alkaline environments promote the precipitation and immobilization of Pb, reducing its mobility and leachability [69]. The mechanism of Pb immobilization in geopolymer is explained elsewhere as the formation of lower soluble Pb compounds like lead silicates (Pb_3_SiO_5_) [9] and encapsulation of metals in the geopolymer matrix [71]. Washing pretreatment further enhances Pb stability by removing readily soluble Pb compounds, resulting in an even lower long-term leachability. Jean-Claude et al. (2002) reported the variation in Pb solubility with different pHs showing the lowest solubility between pH 6.7 and 10.5 [72].

#### 3.5.3. Cu Leachability under Different pH Conditions

At pH 2.88, the initial Cu leaching concentration of GP-FA showed approximately 23 µg/L (Figure 8). Over time, the concentration increased to 204 µg/L. The initial leaching concentration of GP-WFA was around 100 µg/L, which gradually increased to 311 µg/L over the experimental period. This result is likely due to the higher Cu content in GP-WFA. Note that the Cu content of washed MSWIFA was twice as large as that of unwashed MSWIFA (Table 4). At pH 4.2, Cu leaching from GP-FA and GP-WFA was substantially lower compared to pH 2.88. The initial concentration in the GP-FA leaching solution was 19.6 µg/L, decreasing to 7.1 µg/L over time. The Cu leachability of GP-WFA was slightly higher than that of GP-FA. The initial Cu concentration in GP-WFA was 25.2 µg/L and remained between 16 and 20 µg/L over the experimental period. At pH 10, Cu leaching was substantially lower. The initial concentration of GP-FA and GP-WFA was around 14.6 µg/L and 14.8 µg/L, which reduced to 7.7 µg/L and 9.8 µg/L, respectively. This result demonstrates that the heavy metal content and the contact time in the higher pH condition do not significantly affect the leachability of Cu from GP with MSWIFA. Grba et al. (2023) examined the copper ions (Cu^2+^) uptake with a metakaolin-based geopolymer in water and showed that Cu^2+^ was removed by the geopolymer through Cu precipitation as well as incorporation into the geopolymer matrix [73]. This phenomenon substantially occurred in high pH ranges in their study.

#### 3.5.4. Cd Leachability under Different pH Conditions

At pH 2.88, the Cd concentration in the leachate from GP-FA was 1.1 µg/L. Over time, the concentration increased to 16.6 µg/L as shown in Figure 9. The Cd leaching concentration of GP-WFA was 8.7 µg/L initially and gradually increased to 30.5 µg/L over the experimental period. At pH 4.2 and pH 10, Cd leaching was substantially lower compared to pH 2.88. The Cd leaching concentrations from both GP-FA and GP-WFA were below 1.0 µg/L. The lower Cd leachability in higher pH conditions suggested that the formation of insoluble Cd(OH)_2_ and other cadmium hydroxide complexes is the dominant mechanism for the immobilization of Cd [69].

#### 3.5.5. Ni Leachability under Different pH Conditions

The Ni concentration in the leachate from both GP-FA and GP-WFA at pH 2.88 was insignificant and comparable to the Ni leaching concentration of GP-control as shown in Figure 10. At pH 4.2, Ni leaching was lower compared to pH 2.88 but remained lower than the Ni leaching from GP-control. The initial concentration of GP-FA and GP-WFA was 1.0 µg/L and 1.2 µg/L, increasing slightly to 1.8 µg/L and 2.4 µg/L over time. The Ni leaching at pH 10.0 was similar to that at pH 4.2.

#### 3.5.6. Cr Leachability under Different pH Conditions

The leaching behavior of Cr differs from that of other examined metals. As shown in Figure 11, comparing the leachability of Cr in different pH conditions, the leaching concentration of Cr was slightly higher at pH 10. Interestingly, the Cr leachability reduced under acidic conditions (pH 2.88 and pH 4.22) over time but increased under pH 10.0 conditions for GP with both washed and unwashed MSWIFA. Cr mainly exists as Cr(VI) due to the alkaline environment within geopolymer cement [74]. Ji and Pei [74] studied the immobilization efficiency of heavy metals including Cr_2_O_7_^2−^ in a geopolymer and found that the immobilization efficiency of Cr(VI) in geopolymer concrete was relatively low compared to metal cations like Zn^2+^, Pb^2+^, and Cd^2+^. Al-Mashqbeh et al. [75] reported that a geopolymer had a limited immobilization capacity for the encapsulation of Cr_2_O_7_^2−^ due to the formation of a limited number of pores. In their study, the immobilization capacity of metal anions improved by optimizing the Si/Al ratio [75].

## 4. Conclusions

This study investigated the utilization of municipal solid waste incineration fly ash (MSWIFA) in geopolymer concrete, focusing on compressive strength and heavy metal leachability under different environmental conditions. Based on the experimental results, the following key conclusions can be drawn:A Si/Al ratio of 1.5, water/Na ratio of 10, and sand-binder ratio of 0.6 were identified as optimal parameters for achieving the highest compressive strength in geopolymer concrete. The influence of the Si/Al ratio was particularly significant, with a higher Si/Al ratio leading to a stronger geopolymer matrix.The washing pretreatment of MSWIFA significantly enhanced the compressive strength of geopolymer concrete, especially under alkaline conditions. This improvement can be attributed to the removal of soluble salts, which reduces the detrimental impact of contaminants.Geopolymer concrete containing washed MSWIFA (GP-WFA) showed an improved long-term compressive strength compared to unwashed MSWIFA (GP-FA), with GP-WFA reaching a compressive strength comparable to control geopolymer concrete (GP-control) after 16 weeks. Compressive strength was more stable under alkaline conditions (pH 10), while acidic conditions (pH 2.88) led to surface degradation and reduced strength.The leachability of heavy metals such as Zn, Pb, Cu, Cd, and Cr varied significantly with pH levels. Strong acidic conditions (pH 2.88) resulted in a higher leachability, whereas alkaline conditions (pH 10) greatly reduced the leaching of heavy metals. The washing pretreatment increased the leachability of some metals, particularly Zn and Cu, under acidic conditions due to the concentration of these metals in the washed residue.The reduction in heavy metal leachability under alkaline conditions highlights the potential for the safe usage of MSWIFA in construction applications, particularly in environments with higher pH levels.

Further research is recommended to explore the long-term durability of geopolymer concrete with MSWIFA under real-world environmental conditions and to optimize the washing process to minimize metal leachability without compromising compressive strength. Standardization efforts should focus on developing guidelines for the safe implementation of MSWIFA-based geopolymer materials in construction industries.

## Figures and Tables

**Figure 1 materials-17-04609-f001:**
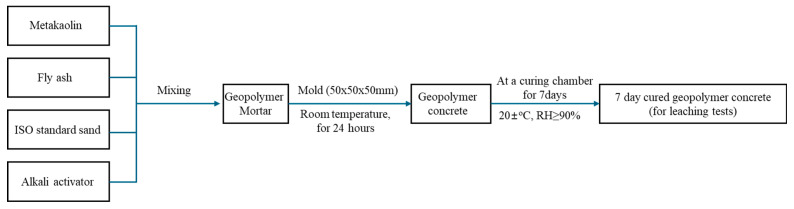
Flow diagram of MSWIFA geopolymer concrete production and leaching tests.

**Figure 2 materials-17-04609-f002:**
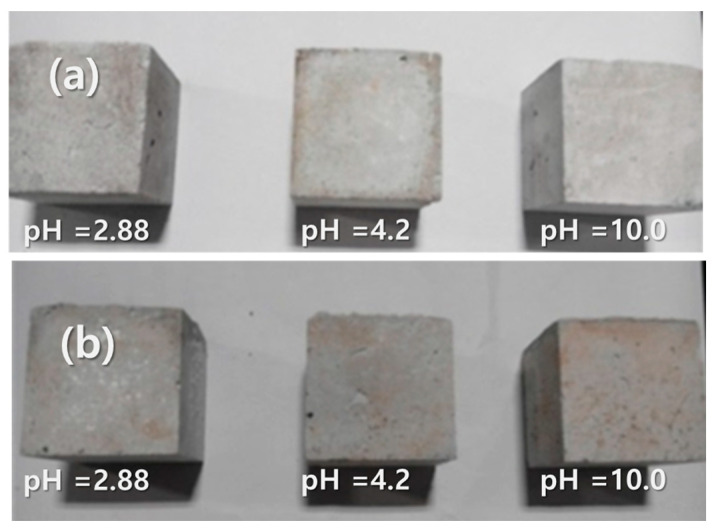
Images of GP-FA before and after the 16-week leaching at different pH levels: (**a**) before exposure to the leaching solution and (**b**) after the 16-week leaching experiment.

**Figure 3 materials-17-04609-f003:**
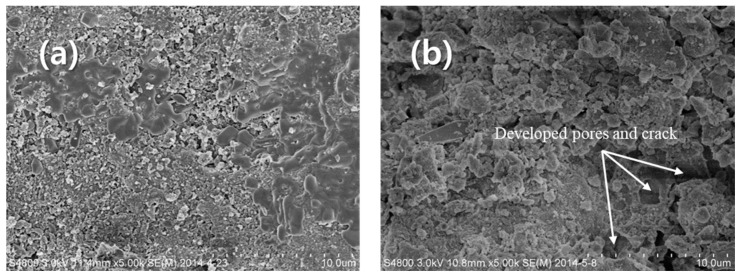
SEM images of GP samples before and after the 16-week leaching at pH 2.88: (**a**) before exposing to the leaching solution and (**b**) after the 16-week leaching experiment.

**Figure 4 materials-17-04609-f004:**
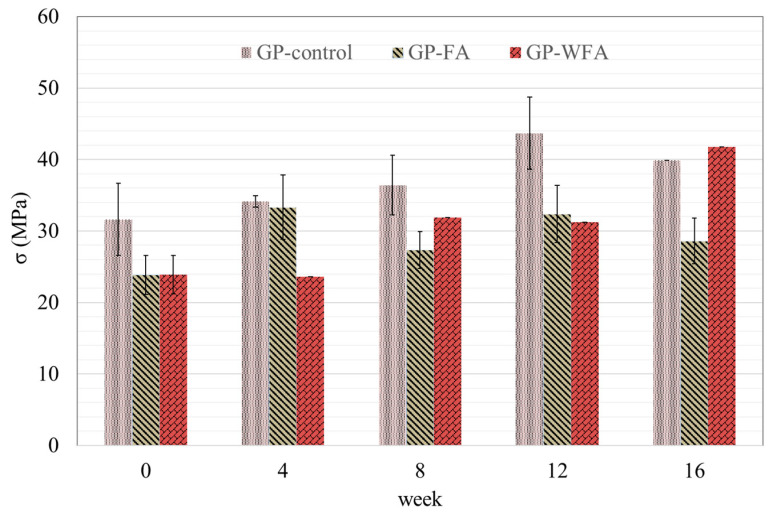
Compressive strength development of geopolymer not exposed to leaching solution.

**Figure 5 materials-17-04609-f005:**
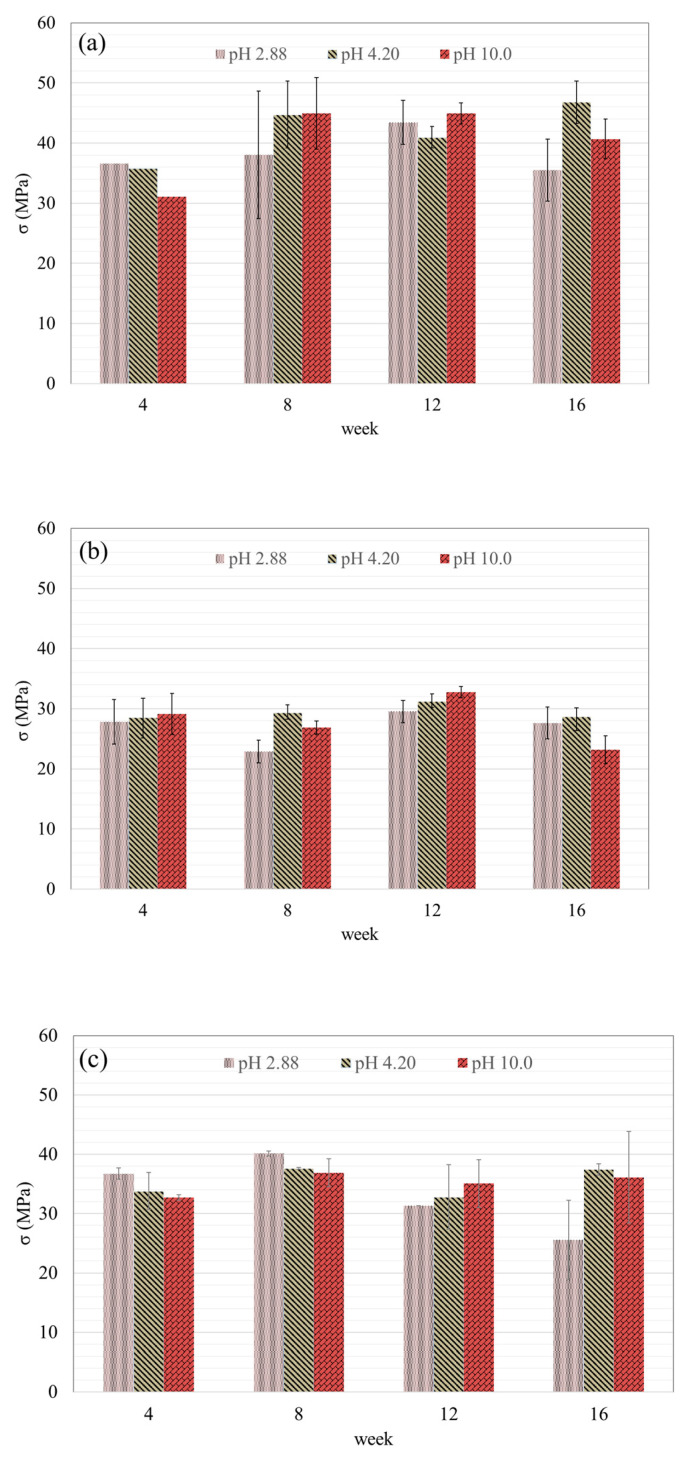
Compressive strength of geopolymer concrete under different pH conditions: (**a**) GP-control, (**b**) GP-FA, and (**c**) GP-WFA.

**Figure 6 materials-17-04609-f006:**
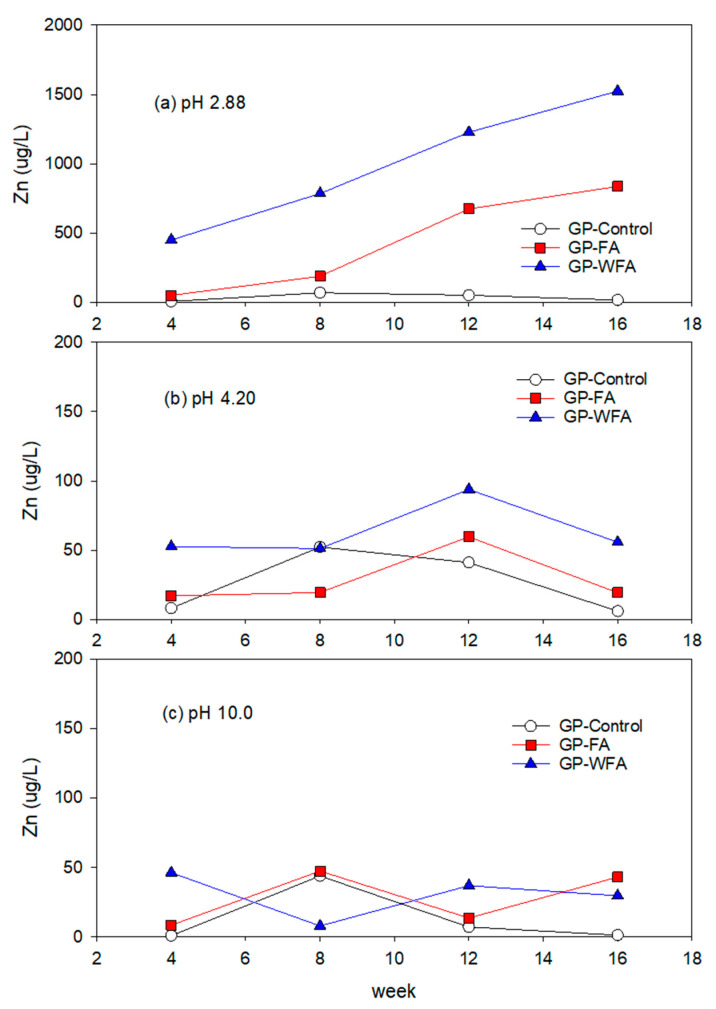
Leaching behavior of Zn at different pH: (**a**) 2.88, (**b**) 4.20, (**c**) 10.0.

**Figure 7 materials-17-04609-f007:**
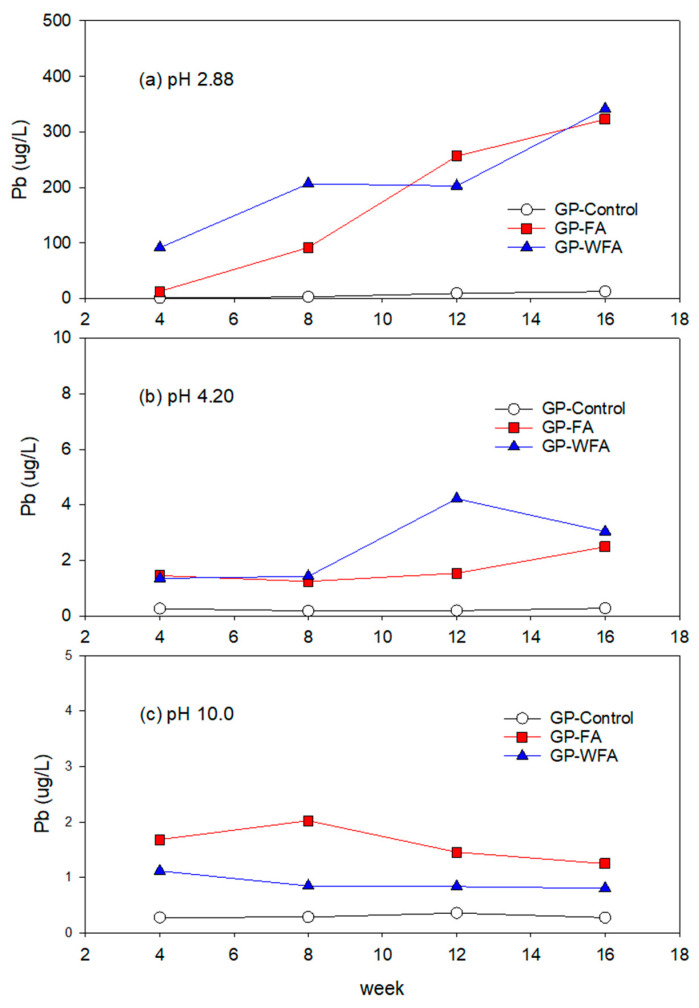
Leaching behavior of Pb at different pH: (**a**) 2.88, (**b**) 4.20, (**c**) 10.0.

**Figure 8 materials-17-04609-f008:**
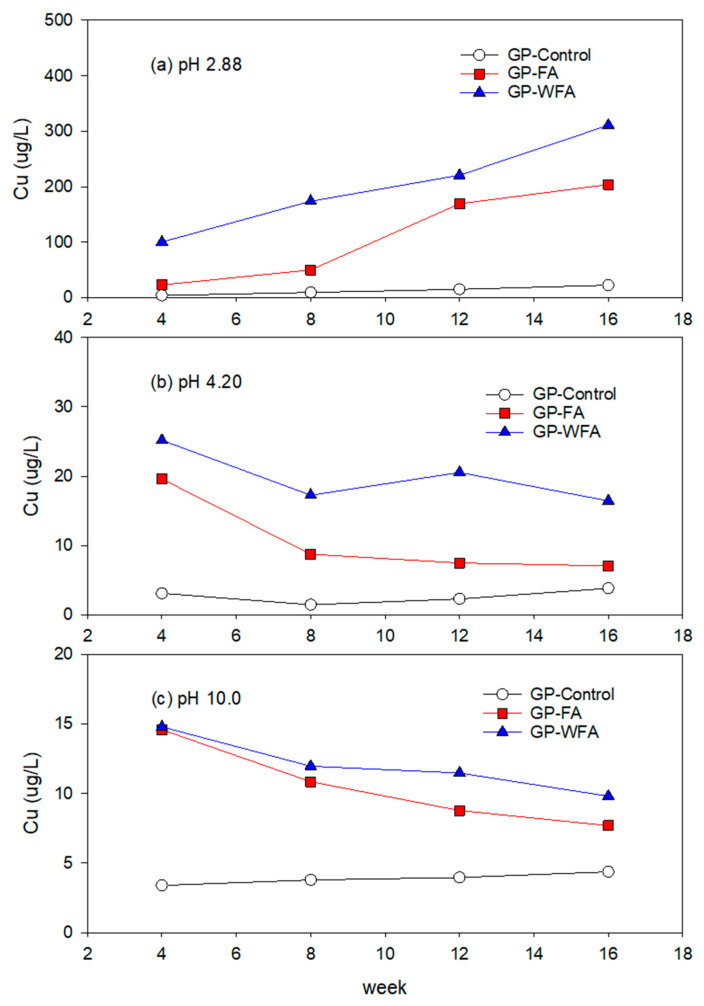
Leaching behavior of Cu at different pH: (**a**) 2.88, (**b**) 4.20, (**c**) 10.0.

**Figure 9 materials-17-04609-f009:**
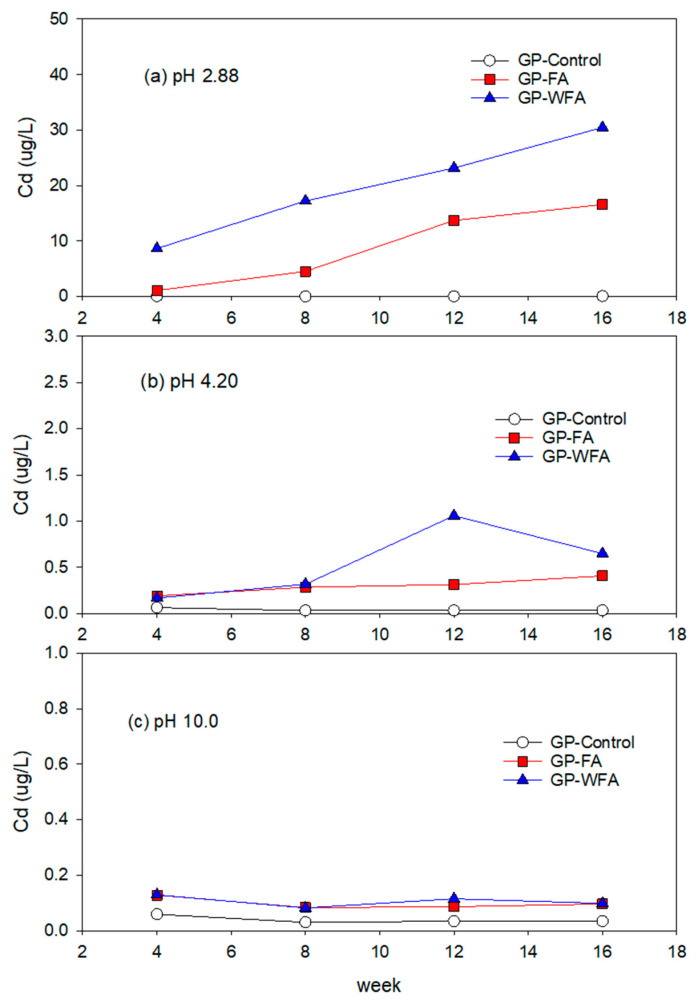
Leaching behavior of Cd at different pH: (**a**) 2.88, (**b**) 4.20, (**c**) 10.0.

**Figure 10 materials-17-04609-f010:**
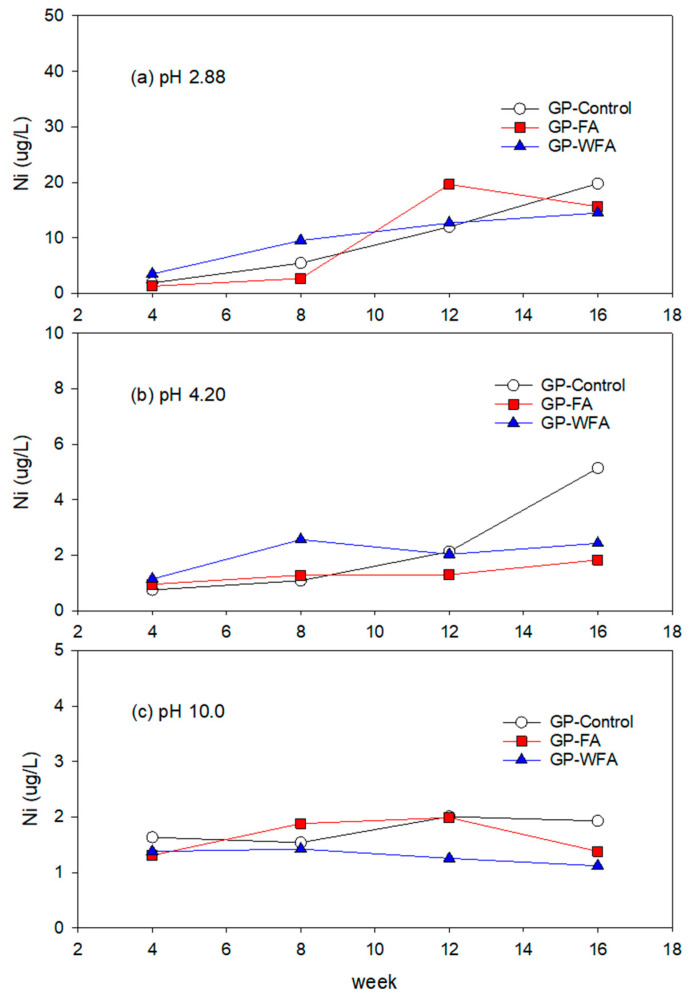
Leaching behavior of Ni at different pH: (**a**) 2.88, (**b**) 4.20, (**c**) 10.0.

**Figure 11 materials-17-04609-f011:**
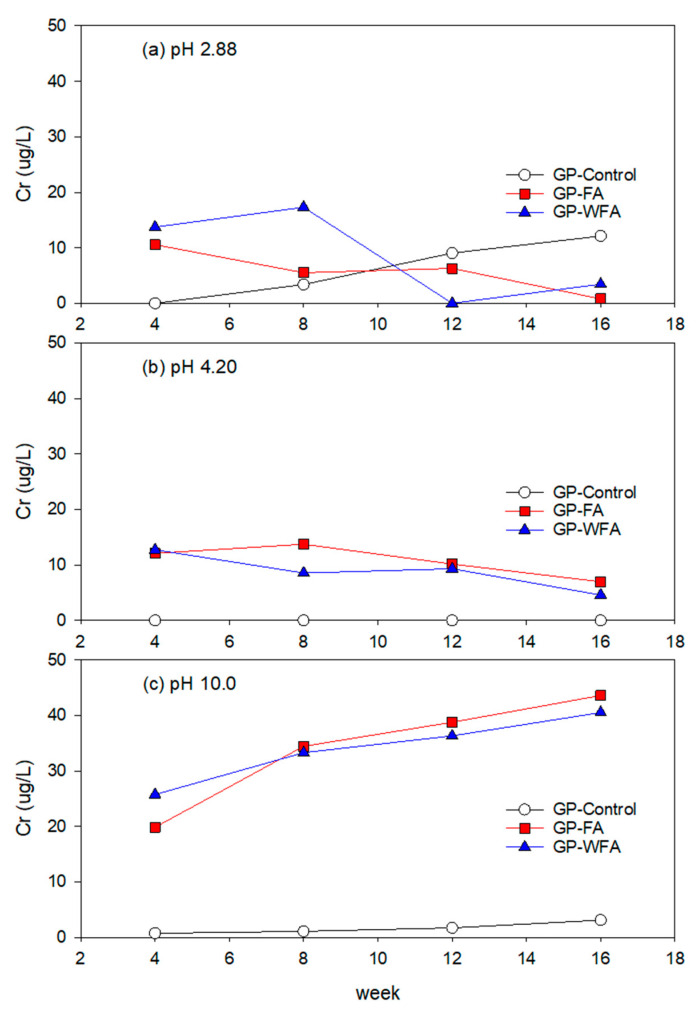
Leaching behavior of Cr at different pH: (**a**) 2.88, (**b**) 4.20, (**c**) 10.0.

**Table 1 materials-17-04609-t001:** Orthogonal experiment design for geopolymer concrete with 0% MSWIFA.

No.	Si/Al Ratio	Water/Na Ratio	Sand/Binder Ratio	Ca(OH)_2_ Addition
1	1.5	7.5	0.6	0
2	1.5	8.5	1.0	0.1
3	1.5	9.5	1.4	0.2
4	1.9	7.5	1.4	0.1
5	1.9	8.5	0.6	0.2
6	1.9	9.5	1.0	0
7	2.3	7.5	1.0	0.2
8	2.3	8.5	1.4	0
9	2.3	9.5	0.6	0.1

**Table 2 materials-17-04609-t002:** Range analysis (MPa).

Level *	Si/Al Ratio	Water/Na Ratio	Sand/Binder Ratio	Ca(OH)_2_ Addition
Level 1	27.4 (±8.9) **	22.5 (±12.8)	19.1 (±16.1)	14.6 (±19.5)
Level 2	10.6 (±5.7)	14.3 (±11.5)	15.7 (±11.9)	14.9 (±10.0)
Level 3	8.6 (±7.7)	9.8 (±8.4)	11.7 (±8.5)	17.1 (±2.5)
Range *	18.79	12.66	7.4	2.49

* Range = Maximum average response—Minimum average response, ** () indicates standard deviation.

**Table 3 materials-17-04609-t003:** Chemical composition and loss of ignition of MSWIFA and metakaolin.

Component (wt. %)		Al_2_O_3_	CaO	Cl	Fe_2_O_3_	K_2_O	MgO	Na_2_O	SiO_2_	SO_3_	LOI
MSWIFA	unwashed	1.0	38.9	9.9	0.5	6.5	0.3	6.9	6.8	7.0	19.6
	washed	1.8	48.2	1.2	0.9	0.7	1.3	1.1	5.6	7.9	26.3
Metakaolin		45.4	0.3	-	-	0.04	-	0.1	51.4	-	1.5

**Table 4 materials-17-04609-t004:** Changes in heavy metal content in unwashed and washed MSWIFA and metakaolin.

Heavy Metal	Unwashed MSWIFA	Washed MSWIFA	Metakaolin
	(mg·kg^−1^)		
Zn	6096	9416	94
Pb	1555	1472	20
Cu	1144	2205	60
Cd	174	341	3
Cr	53	83	21
Ni	436	190	62

## Data Availability

The original contributions presented in the study are included in the article, further inquiries can be directed to the corresponding author.

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
