# Peer review of "Utilization of Municipal Solid Waste Incineration (MSWIFA) in Geopolymer Concrete: A Study on Compressive Strength and Leaching Characteristics"

_materials, 2024, doi:10.3390/ma17184609_

Round 1
Reviewer 1 Report
Comments and Suggestions for Authors
REVIEWER COMMENTS FOR AUTHORS
The manuscript titled “Utilization of Municipal Solid Waste Incineration MSWIFA(MSWIFA) in Geopolymer Cement: A Study on Compressive Strength and Leaching Characteristics” is reviewed and following comments are made.
This article requires major revisions not only for grammatical errors and prefix/suffix mistakes, but also for each technical input. The following suggestions can help improve the article:
1. The title of the manuscript needs to be rephrased and corrected.
2. The Pyramid theory or research reported by Prof. Joseph Davidovits on geopolymers is not highlighted. It is a flaw in the paper.
3. Conclude Abstract section with addition of statistical results.
4. Many topics which are very relevant to geopolymer are missing. Some of them are chemistry behind the geopolymerization, theory of particle packing, Abrams', Bolomey's law, effect of fluid-binder ratio, Variability of source materials. (All this needs to be added in the manuscript) which improves the quality of manuscript.
5. What is the statement of novelty of the research work?! It is missing in the manuscript; it is advised for the authors to add the statement of novelty soon after the introduction section.
6. In table 3 it is must to add Loss of Ignition [LOI] of MSWIFA and Metakaolin.
7. In fig 2 SEM images show the changes or differences by marking between before and after 16-week leaching.
8. The resolution of graphs is poor, replace the graphs with higher resolution.
9. It is advised to add about the effect of Si/Al ratio on the developed geopolymer cement.
10. The results of compressive strength test need to be explained with respect in increase and decrease in percentage values.
11. What was the measure taken to maintain pH throughout the testing?
12. Conclusion needs to be rephrased and rewritten point wise.
13. There are a few typo errors in the entire manuscript, authors need to check carefully using Grammarly or any software’s.
14. It is advised for the authors to refer to the following papers and cover them in the introduction part and for the results and discussion part, wherever applicable, in the current manuscript to improve the overall quality of the manuscript, which was recently published in the Development of Geopolymers using industrial waste materials.
https://doi.org/10.3390/su141710948
https://doi.org/10.1016/j.matpr.2022.04.192
This research manuscript may be incomplete without adding and correcting all these above suggested points.

Reviewer 2 Report
Comments and Suggestions for Authors
The manuscript has interesting results and has the potential to be published at Materials Journal. Some revisions are needed before acceptance, as follows:
1 - Why use the abbreviation MSWIFA twice in the title?
2 - Page 1 - Line 41: This paragraph uses the same idea/information that the paragraph describes above. Please rewrite to not repeat information.
3 - Table 2: This table needs to be clarified. What are the levels? The compressive strength may be placed in this table.
4 - Line 114: What explains the increased metal concentration after washing? The goal of the study was the opposite. Right?
5—Table 3: Note that some important elements in geopolymer, like Si and alkalis, also reduce. It is necessary to discuss these results.
6 - Line 126: It needs to be clarified in Figure 1 the cracks.
7—Line 146: Why is the compressive strength of GP-WFA lower in early ages than that of GP-FA? Use Table 3 and the literature to explain.
8 - Line 264: Explain why this behavior differs from other metals.
9 - Line 282: subscribe +2
10 - The structure of the paper needs to be corrected. Material and methods should be placed before the results and after the introduction.
11—Line 297: Is this topic the same as 3.3? Do you produce two types of geopolymers? Why is the addition of Ca(OH)2 in topic two and not here? What is the composition, brand, and other information about portlandite used?
12—Lines 308 and 309: Please add the velocity of mixing in RPM. More information is needed about alkaline activators. What is the composition?
13—Line 316: Is this experiment used to define the standard composition of geopolymer? If so, this topic should be placed before 3.2.
14 - Line 322, table 1: The compressive strength should be placed in the results section.
Reviewer 3 Report
Comments and Suggestions for Authors
overall good job, some remarks for improvements
1) please specify aim / objective and tasks derived so they correspond to results as fulfilled
2) please improve Figures, hardly being read at the moment, emphasize the main
3) adding some references on landfill mining and potential materials extracted for additives are encouraged
4) some improvements for introduction and conclusions to make it easier for generalists not experts of the field
Comments on the Quality of English Languagestyle and grammar and abbreviation check
Reviewer 4 Report
Comments and Suggestions for Authors
The article titled "Utilization of Municipal Solid Waste Incineration 2 MSWIFA(MSWIFA) in Geopolymer Cement: A Study on Compressive Strength and Leaching Characteristics" (materials-3167094) presents analysis of the potential use of MSWIFA.
The article submitted for review is recommend for acceptation after minor revisions.
1. TITLE, ABSTRACT, INTRODUCTION:
a. Authors repeat MSWIFA in the title.
b. The introduction can be supplemented with research from different parts of the world.
2. MATERIALS AND METHOD
a. Why authors decide put MATERIALS AND METHOD chapter after Results and discussion chapter ?
b. The graphical presantation of sample preparation could be added.
c. Table 1. Why was the orthogonal plan used?
d. line 338 / 343 - what interval was concentration of haevy meatal measured in ?
3. RESULTS / DISCUSSION / CONCLUSION
a. Table 2 - what basis were the parameter ranges selected on?
b. line 125 – there is lack of consistency in the abbreviations used (FA / MSWIFA) ( FA / MSWIFA).
The discussion of the results should be more detailed.
c. Table 4 - why does the content of Pb and Ni decrease after washing and the content of Zn, Cu, Cd, Cr and Ni decrease ?
d. line 130 – why does pH 2.88 have a significant corrosive effect on the surface of GP samples ?
e. lines 139-186 – why does compressive strength of GP samples change (increasing / decreasing) as a function of time ?
f. Figure 5 -9 – the figures are illegible, there is no statistical analysis of results.
g. How does leaching (in different pH) affect the phase composition? What is the mass loss after leaching?
Round 2
Reviewer 1 Report
Comments and Suggestions for Authors
No further revisions required.